# Individual prevention and containment measures in schools in Catalonia, Spain, and community transmission of SARS-CoV-2 after school re-opening

**Sergio Alonso**[1], **Martí Català**[1,2], **Daniel López**[1], **Enric Álvarez-Lacalle**[1], **Iolanda Jordan**[3,4,5], **Juan José García-García**[4,5,6], **Victoria Fumadó**[7], **Carmen Muñoz-Almagro**[4,5,8], **Eduard Gratacós**[9,10,11], **Núria Balanza**[12], **Rosauro Varo**[12,13], **Pere Millat**[12], **Bàrbara Baro**[12], **Sara Ajanovic**[12,13], **Sara Arias**[12], **Joana Claverol**[4,14], **Mariona Fernández de Sevilla**[4,5,6], **Elisenda Bonet-Carne**[9,10,15], **Aleix Garcia-Miquel**[9], **Ermengol Coma**[16], **Manuel Medina-Peralta**[16], **Francesc Fina**[16], **Clara Prats**[1‡], **Quique Bassat**[5,6,12,13,17‡]*

1 Department of Physics, Universitat Politècnica de Catalunya, Barcelona, Spain, 2 Comparative Medicine and Bioimage Centre of Catalonia (CMCiB), Fundació Institut d'Investigació en Ciències de la Salut Germans Trias i Pujol, Badalona, Catalonia, Spain, 3 Paediatric Intensive Care Unit, Hospital Sant Joan de Déu, University of Barcelona, Barcelona, Spain, 4 Institut de Recerca Sant Joan de Déu, University of Barcelona, Barcelona, Spain, 5 Consorcio de Investigación Biomédica en Red de Epidemiología y Salud Pública (CIBERESP), Madrid, Spain, 6 Pediatrics Department, Hospital Sant Joan de Déu, Universitat de Barcelona, Esplugues, Barcelona, Spain, 7 Infectious Diseases Department, Hospital Sant Joan de Déu, Barcelona, Spain, 8 Department of Medicine, Universitat Internacional de Catalunya, Barcelona, Spain, 9 BCNatal | Fetal Medicine Research Center (Hospital Clínic and Hospital Sant Joan de Déu), University of Barcelona, Barcelona, Spain, 10 Institut d'Investigacions Biomèdiques August Pi i Sunyer (IDIBAPS), Barcelona, Spain, 11 Center for Biomedical Research on Rare Diseases (CIBER-ER), Madrid, Spain, 12 ISGlobal, Hospital Clínic—Universitat de Barcelona, Barcelona, Spain, 13 Centro de Investigação em Saúde de Manhiça, Maputo, Mozambique, 14 Fundació Sant Joan de Déu, Barcelona, Spain, 15 Universitat Politècnica de Catalunya BarcelonaTech, Barcelona, Spain, 16 Sistema d'Informació dels Serveis d'Atenció Primària (SISAP), Institut Català de la Salut, Barcelona, Spain, 17 ICREA, Catalan Institution for Research and Advanced Studies, Barcelona, Spain

‡ These authors contributed equally to this work and are joint senior authors on this work.
* quique.bassat@isglobal.org

**Data Availability Statement:** Availability of data and material (data transparency) Epidemiological data used for the assessment of 10-day cumulative incidence and organized by age and positivity are

## Abstract

### Background

Despite their clear lesser vulnerability to COVID-19, the extent by which children are susceptible to getting infected by SARS-CoV-2 and their capacity to transmit the infection to other people remains inadequately characterized. We aimed to evaluate the role of school reopening and the preventive strategies in place at schools in terms of overall risk for children and community transmission, by comparing transmission rates in children as detected by a COVID-19 surveillance platform in place in Catalonian Schools to the incidence at the community level.

### Methods and findings

Infections detected in Catalan schools during the entire first trimester of classes (September-December 2020) were analysed and compared with the ongoing community

public and can be found in: • https://dadescovid.
cat/descarregues?lang=eng • https://dadescovid.
cat/static/csv/catalunya_diari.zip Data regarding
positive cases and confined groups in schools are
also publicly available and can be found at: •
https://analisi.transparenciacatalunya.cat/en/
Educaci-/Dades-COVID-19-als-centres-educatius/
fk8v-uqfv Aggregated data of secondary cases per
group in primary and secondary schools were
provided under request by Traçacovid project from
the Education Department of the Catalan
Government and are shown in Supplementary
Table 1. These are third party data, which are, with
the exception of aggregated data of secondary
cases per group in primary and secondary schools,
all freely available from the abovementioned
webpages. Authors did not have special privileges
in relation to obtaining access to those data.

**Funding:** Funding The components of the analysis
drawn from the different KIDS Corona platform
have been funded by Stavros Niarchos Foundation
(SNF), Banco Santander and other private donors
of Kidscorona. We also acknowledge funding from
La Caixa Foundation (ID 100010434), under
agreement LCF/PR/GN17/50300003; and funding
from Ministerio de Ciencia, Innovación y
Universidades and FEDER, with the project
PGC2018-095456-B-I00. This work has been also
partially funded by the European Commission - DG
Communications Networks, Content and
Technology through the contract LC-01485746.
ISGlobal receives support from the Spanish
Ministry of Science and Innovation through the
"Centro de Excelencia Severo Ochoa 2019-2023"
Program (CEX2018-000806-S), and support from
the Generalitat de Catalunya through the CERCA
Program. CISM is supported by the Government of
Mozambique and the Spanish Agency for
International Development (AECID). NB is
supported by an FPU predoctoral fellowship from
the Spanish Ministry of Universities (FPU18/
04260). BB is a Beatriu de Pinós postdoctoral
fellow granted by the Government of Catalonia's
Secretariat for Universities and Research, and by
Marie Sklodowska-Curie Actions COFUND
Programme (BP3, 801370). Role of the funding
source The funders had no role in the interpretation
of the data or in the writing up of the manuscript.

**Competing interests:** The authors have declared
that no competing interests exist.

transmission and with the modelled predicted number of infections. There were 30.486 infections (2.12%) documented among the *circa* 1.5M pupils, with cases detected in 54.0% and 97.5% of the primary and secondary centres, respectively. During the entire first term, the proportion of "bubble groups" (stable groups of children doing activities together) that were forced to undergo confinement ranged between 1 and 5%, with scarce evidence of substantial intraschool transmission in the form of chains of infections, and with ~75% of all detected infections not leading to secondary cases. Mathematical models were also used to evaluate the effect of different parameters related to the defined preventive strategies (size of the bubble group, number of days of confinement required by contacts of an index case). The effective reproduction number inside the bubble groups in schools (R*), defined as the average number of schoolmates infected by each primary case within the bubble, was calculated, yielding a value of 0.35 for primary schools and 0.55 for secondary schools, and compared with the outcomes of the mathematical model, implying decreased transmissibility for children in the context of the applied measures. Relative homogenized monthly cumulative incidence ($r_{CI_{hom,j}}$) was assessed to compare the epidemiological dynamics among different age groups and this analysis suggested the limited impact of infections in school-aged children in the context of the overall community incidence.

## Conclusions

During the fall of 2020, SARS-CoV-2 infections and COVID-19 cases detected in Catalan schools closely mirrored the underlying community transmission from the neighbourhoods where they were set and maintaining schools open appeared to be safe irrespective of underlying community transmission. Preventive measures in place in those schools appeared to be working for the early detection and rapid containment of transmission and should be maintained for the adequate and safe functioning of normal academic and face-to-face school activities.

## Introduction

Much has been discussed in the past months on the role of children in the current COVID-19 pandemic, and their contribution to the overall transmission at the community level [1]. While it appears now clear that children are amongst the least vulnerable population group in society, given the limited clinical expression of their infections [2,3], many uncertainties still exist in terms of their susceptibility to infection [4], and their capacity to infect and spread SARS-CoV-2 [5–7]. Indeed, while some outbreaks (defined as chains of infections affecting at least three individuals) in nurseries, schools, and summer camps have been reported [8–10], it is still debatable the extent to which children of varying ages can be effective drivers of super shedding events [6]. Understanding transmission potential from children is of paramount importance to better design prevention and containment measures in settings where large gatherings of children occur, such as nurseries, primary or secondary schools, or extra-academic activities.

During the initial phases of the pandemic, and possibly influenced by the hitherto existing general understanding of respiratory viral infections in children, schools were considered high-risk settings and thus rapidly closed, with the belief that this would contribute

substantially to the containment of the epidemic. Although some authors defended, based on modelling approaches, the uncertain effectiveness of such measures [11], advocating for an agile reopening of schools [12–14], others warned of the potentially deleterious effects of children returning to classes, recommending prudence [15–17]. The reality is that schools remained closed in most countries, and children often underwent considerably harsh lockdowns. In Spain, this meant that children remained confined at home for over 3 months, with the inherent physical and mental health risks of such situation [18].

Although scant, available evidence during the first year of the pandemic suggested that secondary transmission in schools was low [19–21] and occurred comparatively less frequently than among other group activities involving adults [6,22,23], and that therefore, schools contributed very modestly to the overall transmission and burden of COVID-19. Successive waves of the pandemic and the emergence of variants with higher transmissibility potential [24,25] did re-ignite this debate [17,26,27], triggering the closing down of schools in many European countries during the academic year 2020–21.

We hereby present a variety of analyses exploring two important subjects regarding childhood SARS-CoV-2 infections in Catalonia, one of the most affected regions of Spain, whereby data were gathered during the fall of the year 2020, one of the highest incidence periods of the pandemic. On one hand, we tried to model and critically assess some of the measures implemented to prevent secondary transmission in school settings. On the other hand, we compared the anticipated modelled consequences of reopening schools in terms of their impact on overall transmission, *vs*. what really was documented to occur during the first trimester after pupils returned to fully face-to-face education [19]. The intention is to demonstrate the limited role of childhood intra-school transmission in relation to overall community transmission when proper epidemiological surveillance and recommendations at schools are in place.

## Methods

### Study setting and design

This study was conducted in Catalonia, one of the 17 autonomous communities of Spain. According to the Catalan Institute of Statistics (IDESCAT; https://www.idescat.cat), the region had a population of 7,727,029 inhabitants in fall of the year 2020, with 1,177,433 (15.2%) being younger than 15 years of age. In Spain, children younger than 6 years old attend pre-primary schools, those between 6 and 11 attend primary schools and 12-age and older children and youth go to the secondary centres. Within the Catalan region, and for the year 2019, there were an estimated 3,967 pre-primary and primary schools (2,662 public, 1,305 private) and 1,105 secondary schools (587 public, 518 private). There were 58 additional special schools, where music or other activities are conducted. Thus, we estimated a total of 5130 schools in Catalonia. The department of education of the local Government of Catalonia estimates that those schools included up to 1,436,680 pupils and 115,413 teachers and other staff members. COVID-19 cases in schools affecting both children and adults are updated daily through the project *Traçacovid* (http://educacio.gencat.cat/ca/actualitat/escolasegura/tracacovid/).

### Prevention and containment measures in catalan schools

On account of the moderate-to-high underlying community transmission in most of the Catalan territory as of the end of August 2020, Catalan authorities proposed a series of prevention and containment measures for schools with the objective to 1) minimize individual risk of pupils and adult staff members (teachers and other staff); 2) ensure rapid detection and subsequent isolation of any positive cases, and 3) isolate and screen close contacts of those positive

cases. These measures were devised to limit the entry of SARS-CoV-2 carriers in schools or to rapidly contain any potential source of outbreaks. By doing this, the idea was to demonstrate that an agile prevention, detection, and containment strategy would curtail intra-school transmission, and therefore not further impact wider community transmission. Individual prevention measures included frequent hand hygiene, the compulsory use of face masks at all times (except during lunch breaks) for anyone aged 6 years and older (or from primary school onwards), physical distancing, and promoting ventilation and outdoor activities when possible. The containment strategy included the stringent recommendation that children (and adults) with COVID-19 compatible symptoms were not to attend school and the establishment within the schools of stable social contact groups (termed "bubbles"). Bubbles should include a "manageable" group of children (ideally smaller than the normal class size which is typically ~30) and their teacher(s), and individuals within a bubble should not establish contact inside the school with members of other bubbles. Bubbles would also facilitate rapid reaction to any SARS-CoV-2 positive case, including a mandatory 14-day (subsequently reduced to 10-day) quarantine for all bubble members, and the recommendation for viral molecular screening to all of them. Demonstration of transmission occurring in different bubbles within the same school could entail school closure, should the health authorities deem it adequate. Although originally planned, no mass screening and testing of teachers or pupils was conducted during the first trimester.

## Infection and disease surveillance in Catalonia

Real data from incident new SARS-CoV-2 infections in Catalonia, disaggregated by age groups, were gathered from the SISAP information systems (*Sistema d'Informació dels Serveis d'Atenció Primària*). These data which can be freely downloaded from *DadesCovid* portal (https://dadescovid.cat/descarregues?lang=eng) were used to contrast our simulations and predictions, once the whole first term (September 1st 2020-December 31st, i.e., ~4 months) of school activities had elapsed, and specifically putting them in context of the incidence trends witnessed throughout the pandemic (including also the subsequent 3-week holiday break).

Given that the reported incidence is affected by the diagnostic effort, the higher number of tests performed in the scholar context can bias the comparison between children and other age groups, or with that at the community level (see S1 Fig). For this specific purpose, we defined the relative homogenized cumulative incidence ($r_{CI_{hom,j}}$), a measure that combines the monthly cumulative incidence and the positivity in each age group with regards to the monthly cumulative incidence and the positivity in the general population. We assumed that positivity (i.e percentage of PCR ad rapid antigenic tests that gives rise to a positive result) is an indicator of the diagnostic capacity. Then we obtain an indicator that allows for a better comparison between groups, beyond the simple ratio between incidences:

$$r_{CI_{hom,j}}, = \frac{CI_j \cdot Pos_j}{CI_{pop} \cdot Pos_{pop}}$$

where $CI_j$ is the monthly cumulative incidence in the age group $j$ (i.e., monthly reported cases per $10^5$ persons in this group), and $Pos_j$ is the corresponding positivity rate during this month in this age group. The same variables are calculated for the general population (*pop*). Although this indicator does not aim to evaluate the real incidence, it provides a way to describe the epidemic dynamics and the relative incidence in each age group taking the diagnosis effort into account.

## Hypothesis

We worked under the overarching hypothesis that schools would accurately translate the concurrent transmission in their neighbourhoods/areas, that is, that if a school was located in a high incidence area, cases would be detected in the school with a similar incidence. Conversely, if the school included individuals from low-incidence settings, few cases (or none) would be detected. We also hypothesized that provided schools would adequately implement and follow the pre-established recommendations, incident cases would be adequately detected (particularly if symptomatic), their close contacts screened, and the episode with outbreak potential rapidly contained by isolating the entire bubble. By doing this, intra-school transmission would be limited, and schools could remain open without fuelling overall community transmission.

## Mathematical model

We employed a stochastic computational model to be used as a platform for numerical simulations. In the model, we considered N students per bubble in a school of Nc bubble groups. We performed a temporal evolution of the virtual bubble group for 50 days. We considered two infection pathways: first, a student may be infected outside of the school with a probability proportional to the local incidence (which for our numerical simulations are kept constant), and second, a student could also be infected inside the school by an infectious pupil of the same bubble. In such case, the probability of contagion depends on the cycle of the infection mimicking the probability density function suggested by McAloon *et al* [28]. At the 5th day of the infection, we evaluated a certain probability that the child would be detected by contact tracing or on account of the presence of symptoms, leading in such case to the whole group going into quarantine. Globally, both methods of detection lead to a 70% detection rate of the infected students. Asymptomatic population is estimated to be around 35% of the cases [29], therefore if the infection of the children is inside the house 65% of the cases are symptomatic and likely to be detected. Other cases are symptomatic children (50%) coming from an asymptomatic infection or directly from the school [30]. For more information about the numerical simulations, see the Supplemental Material in S1 File.

We systematically studied the system, performing 100 runs for each set of 5,130 schools, whereby each simulation corresponded to a scholar year (250 days). We kept constant the community active cases during the simulations, assuming therefore R = 1. Once a child is detected because he/she presents symptoms (30% of the cases) or because of contact tracing from outside the school (40% of the cases), the whole class goes to quarantine during 5, 10 or 14 days, depending on the simulation.

Following the Danish school's example, the concept of bubbles was to be a central component of the containment strategy. We were particularly interested in assessing whether the size (in terms of the total number of persons included) of the bubbles had implications in terms of the risk of intra-school transmission. To evaluate the relevance of the size of the bubble and of the duration of a quarantine imposed on contacts of index cases, we performed a set of virtual experiments using the stochastic computational model. We simulated different propagation scenarios, with the probability of infection derived from successive interactions within the bubble of infected (irrespective of whether symptomatic or not) individuals.

Additionally, we used our models to predict the expected number of confined groups in schools in Catalonia, and we compared them with the real number of classes confined according to the department of education. Finally, we critically evaluated the role of schools and children attending them in the incidence trends and overall transmission of SARS-CoV-2 in Catalonia, by assessing the relative homogenized incidence in the 0–9 and 10–19 age groups

(age breakdown as proposed by Catalan authorities) with regards to the general population in Catalonia.

## School's effective reproduction number

The effective reproduction number (R) is generally used to estimate transmission rates of infectious diseases in the general population [31]. Previous studies [32] evaluated transmission inside households using a household effective reproduction number (R*) that was defined as the average number of persons infected by each primary case inside households. This index was adapted to evaluate the transmission inside summer schools by Alonso *et al* [*19]* and Jordan *et al* [33] defining the effective reproduction number in summer schools as the average number of summer school mates infected by each primary case (i.e., the number of secondary infections per index case, $S_i$). In the same way, we defined R* as the effective reproduction number, or average number of children infected by the index case, inside the bubble groups, where $N_{index}$ corresponds to the total number of index cases found:

$$R^* = \frac{\sum_i S_i}{N_{index}}$$

Given that contagious students can also infect close contacts outside the school, it is expected that R*<R. The relationship between R* and R is not straightforward and any comparison must be carried out with caution, because of the different ways to calculate them. Nevertheless, both of them are indexes that aim to assess the average number of contagions that are generated by each single case. Besides, both indexes must be placed in the appropriate time framework, since R is measuring the general transmission without time constraints but R* measures a transmission that is restricted to the time when students go to the school. As a first approximation, we can therefore assume that the relationship between R* and R depends on the weekly time ratio spent inside schools. If children spend 30% of the time inside schools, R*≈0.3·R. This is, of course, a rough approximation that should be considered as an order of magnitude. The real ratio between both effective reproduction numbers will depend not only on the time ratio but also on the kind of social interactions in each context and the characteristics of the surrounding environment (mainly indoors vs outdoors, level of ventilation, and type of social-contact network).

## Ethical issues

All methods were carried out in accordance with relevant guidelines and regulations. All activities requiring human interaction, follow-up or human sampling and testing, at the basis of the calculations of some of the parameters necessary for our numerical simulations, were conducted following protocols approved by the Clinical Research Ethics Committee of the Hospital Sant Joan de Déu de Barcelona (Spain). School and "bubble group" data were analysed, but no individual-level student/pupil/staff data were used, and no informed consent was sought for these analyses.

## Results

### Effect of the size of the bubble group

We considered a total of 5130 schools, with an average of 300 students each. We evaluated five scenarios changing the ratio of students per bubble group: 10, 15, 20, 25, and 30 students per bubble-group, and employed a constant 14-day cumulative incidence of 250 cases/100,000 inhabitants in the community to account for active cases, considering that the initial incidence

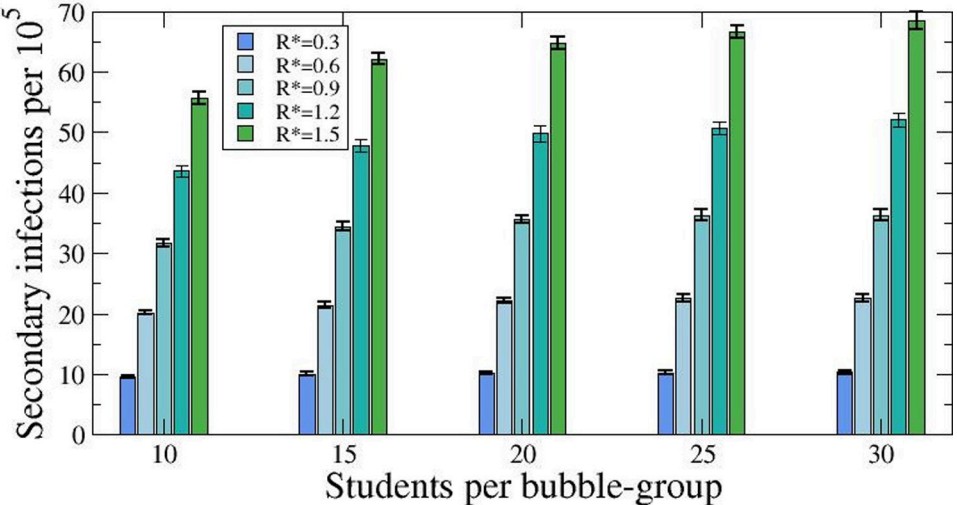

**Fig 1. Variation of the number of secondary infections per 10,000 students according to the size of bubble groups.**
Number of secondary cases obtained from the numerical simulations by varying values of $R^*$ and numbers of scholars forming the bubble groups after the detection of the first infected student. Simulations done keeping $A_{14} = 250$ cases per 100,000 inhabitants, and the total number of students fixed to 1,436,680 ($A_{14}$ = Cumulative incidence over 14 days).

inside the schools is independent on the size of the bubble groups. Assuming that the incidence outside of the school is constant during the time of the simulation, we obtain a continuous flow of infected individuals: about 8,750, corresponding to 1.7 individuals per school during the 50 days of the simulation. This value is independent of the dynamics inside of the school. For each scenario, we simulated five different levels of propagation inside the bubble groups, with an $R^*$ ranging from 0.3 to 1.5, and we assessed the secondary cases inside bubble groups.

Fig 1 illustrates that the number of secondary cases inside the bubble groups increases with the effective reproduction number, as expected. Nevertheless, the role of the size of the bubble group seems relevant only in scenarios of high transmission rate. While for small infection probabilities inside the school the differences are minimal, the ratio becomes more important for large values of $R^*$.

## Effect of the duration and type of quarantine recommended to contacts of index cases

Fig 2 shows the results for different propagation probability inside the school, with an $R^*$ ranging from 0.3 to 1.5. There is a clear inverse correlation between the number of secondary cases with longer quarantine periods up to 10 days, for all values of $R^*$. However, the additional decrease in secondary infections remains modest between 10 and 14 days, even in the context of increasing values of $R^*$.

## Predicted *vs.* actual confined groups after school reopening

As of December 22nd, 2020 (end of first school term), 14 weeks after the opening of the first schools in Catalonia, the number of infected children attending schools reported was 30,486 (2.12% of all pupils), and the number of adult staff infected was 3,143 (1.96%). These infections were detected in 3,226 (63%) schools and led to the confinement of ~1–5% of the bubble groups, and the temporary closing of few schools (typically less than 5 at any given time,

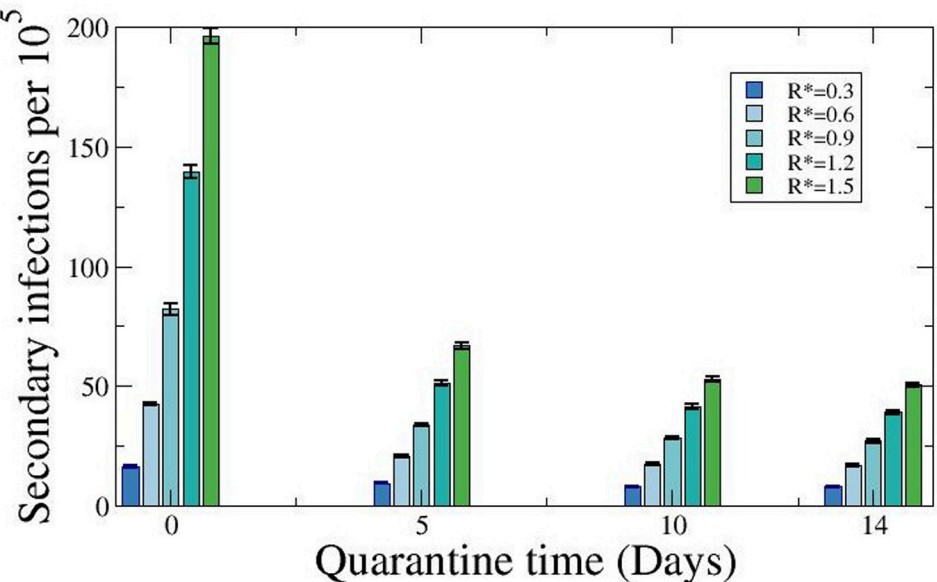

**Fig 2. Variation of the number of secondary infections per 10,000 students according to the days of quarantine.**
Number of secondary cases obtained from the numerical simulations tuning the values of $R^*$ and varying the number of quarantine days after the detection of the first infected individual. Simulations done keeping $A_{14} = 250$ cases per $10^5$ inhabitants, and the total number of groups of 20 students fixed to 72,000 and therefore the total number of students fixed to 1,440,000.

<0.1%). At any stage during those 14 weeks, the proportion of bubble groups affected out of the total number of groups in Catalonia ranged between 1–5%. The distribution of secondary cases per bubble group in school outbreaks inside primary and secondary schools is described in **S1 Table**. We observe differences between incidence in primary and secondary schools. This agrees with observations at the community level during the same period in Catalonia, where differences in terms of the distribution of cases among the different paediatric age groups was found, with older children (aged 12 and above) being more affected than the younger ones (p<0.001) [27].

We can estimate the number of confined groups (EsN) given the incidence of COVID-19 in the whole of the population because the influx of infected students usually comes from the exterior of the school and depends on the fraction of detected students and is proportional to the percentage of time outside the school (if students do not leave the school there are no infections):

$$EsN_{10} = (1 - \eta)\xi \frac{N_G \cdot N_s \cdot A_{10}}{100.000}$$

For this estimate, we used the incidence accumulated over 10 days ($A_{10}$) because the confined groups are accumulated for this time according to the duration of quarantine. The rest of parameters are the average number of students per class ($N_s = 20$) the total number of bubble-groups ($N_G = 72000$), the probability of detection ($\xi$) and the fraction of time spent with the members of the bubble-group ($\eta$). We gave the combination of parameters $(1-\eta)\xi = 0.35$, which may imply that the effective time spent at school is larger than 30% and closer to 50% (children in parks, social activities of teenagers, and others accounting as in class) or that the probability of detection is lower than the 70%, for example 50%. Such parameters may depend on the stress of the sanitary system and the confinement situation of the region.

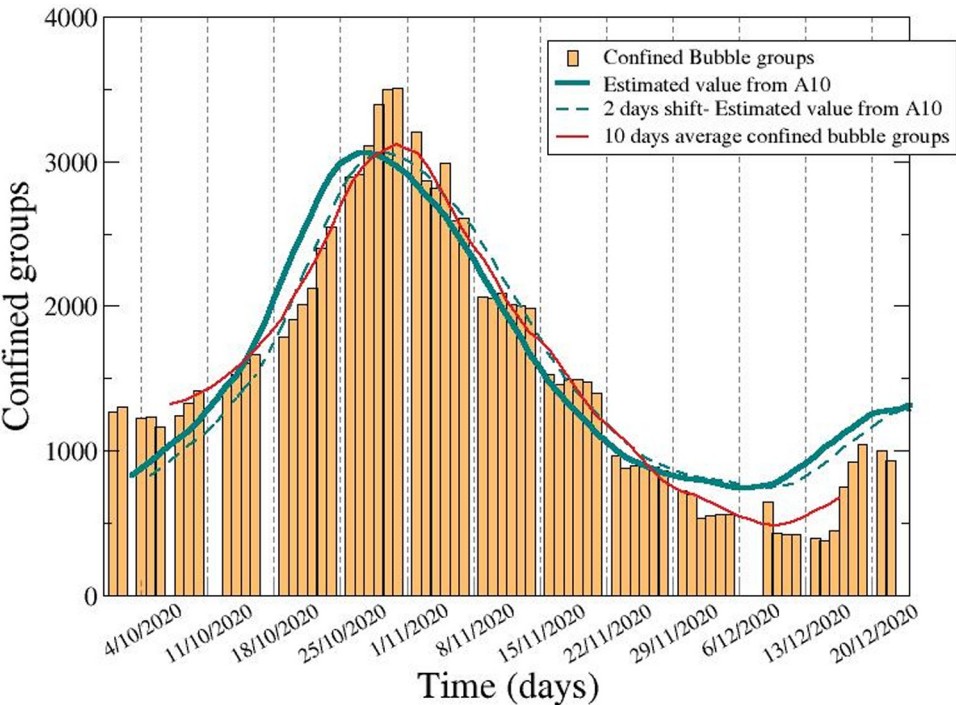

**Fig 3. Evolution of the number of confined groups in schools in Catalonia compared with the estimations based on the incidence.** Temporal evolution of the confined groups in Catalonia from 1st October to 31st December 2020 and the corresponding 7-days moving average (red solid line). Comparison with the estimations proportional to the 7-days moving average of the 10-day cumulative incidence in Catalonia using (1-η) = 0.35 (A10, green solid line), and with the same curve with a 2-day shift (green dashed line) to highlight the delay between the A10 and the actual dynamics of the number of confined groups.

Fig 3 describes the evolution of the number of confined groups in Catalan schools and compares it to the estimations based on the incidence derived from the previous equation. Although during the first days the numbers of groups predicted were lower than actual numbers confined, predictions and reality converged in a very accurate way from day 5 onwards, giving a clear idea of the correlation between cases detected at schools and community transmission in the area where that school was. There is a two-day shift of between the value of the $A_{10}$ in comparison with the confinement of the bubble groups which may be related to the earlier detection inside the house than in the schools, on account of the little and unspecific nature of COVID-19 symptoms in children.

## Estimation of the effective reproductive number inside schools in Catalonia, R*

The increase in secondary cases due to a larger value of R* increases the probability of detection. If there is a larger number of infected individuals within a bubble, the chances of any of them being detected -and thus the group confined- increase either due to the appearance of clinical symptomatology or because of contact tracing from outside.

The number of students infected outside is independent of the probability of infection inside the school, and therefore is constant apart from the fluctuations due to the proper stochasticity of the method. The number of children infected inside of the school depends on R*, which informs about the average number of infected people based on the number of infected students arriving in the school from the outside. As a first approach, we can estimate how

many people can be infected from the index cases and compare it to the value of infected people (which may be actually originated from secondary cases). At least a 40%-50% of the possible infections are avoided due to the use of quarantines, assuming that quarantine conditions are met, i.e., the children are perfectly isolated. When comparing non-quarantine condition cases (0 days quarantine values in Fig 2) with quarantine of 10 days in the same figure, there is an overall reduction for all the values of $R^*$.

Next, we examined with stochastic simulations the number of infected students inside the group before the group goes into quarantine. Once a positive case is found, the whole bubble group is screened and possible transmission chains inside the school can be identified if additional positive cases are found. These diagnoses occur during the application of the protocol for quarantining a group. This information was available from quarantined groups in Catalonia during fall 2020, extracted from the local government documents. From the data summarized in S1 Table we show the fraction of the number of secondary cases per each index case of this period. From these data, we calculated the $R^* = 0.35$ for primary schools and $R^* = 0.55$ for secondary schools and compared these values with our numerical simulations.

We explored different values of $R^*$ and compared the simulation outcome with the data for primary and secondary schools provided by the Catalan government. In Fig 4 this comparison is shown for different values of the probability of infection inside of the bubble group (from $R^* = 0.15$ to $R^* = 0.90$), for the primary and the secondary schools in Fig 4A and 4B. The probability of infection outside the school remains $R = 1$ (i.e., constant incidence). The values are representative of different situations. The scenario with $R^* = 0$ was not explored because it would provide just one single case per quarantined group. We observe that our model could approximately fit the results from the collected data from schools (Fig 4). The model predicted different values of $R^*$ for primary and secondary schools. The differences between the results from the simulations shown in Fig 4A and 4B and the data reported by the schools are minimal for values of $R^*$ in the simulations similar to the values of $R^*$ obtained from the schools, see the mean square displacement shown in Fig 4C and 4D. Such values of $R^*$ correctly reproduced the fraction of outbreaks with low number of secondary infections, however large numbers of secondary infections may need larger values of $R^*$.

For lower values of $R^*$, the fraction of single infected students is too large in comparison with the reported data and, moreover, the fraction of secondary cases underestimated. On the other hand, for higher values of $R^*$ the fraction of single and secondary cases is substantially different than the one observed by the government institutions (see Fig 4E and 4F). Depending on the value of $R^*$ there is a different distribution and number of groups where there is no propagation, and the index case remaining the unique case decreases for large values of $R$ (Fig 4E). Furthermore, the probability of a single person infected by the index case increases with $R^*$ (Fig 4F). Both probabilities are compared in Fig 4E and 4F with the values of the distribution of cases in primary and secondary schools, being closely related to the values of $R^* = 0.35$ and $R^* = 0.55$ observed in the data shown in S1 Table.

An important proportion of school outbreaks in primary (79%) and secondary-High schools (71%) do not seem to lead to any secondary infections. Groups with more than six infected cases, which diverge from the expected decay for the outbreaks with low cases are shown in Fig 4A and 4B. Such cases are not reproduced from the model where interactions are randomly assumed, meaning larger outbreaks are probably associated to superspreading situations. This type of high clustering where a disproportionally large number of cases are generated by a few cases is a rather well established property of the secondary attack rate structure in other environments like households [34].

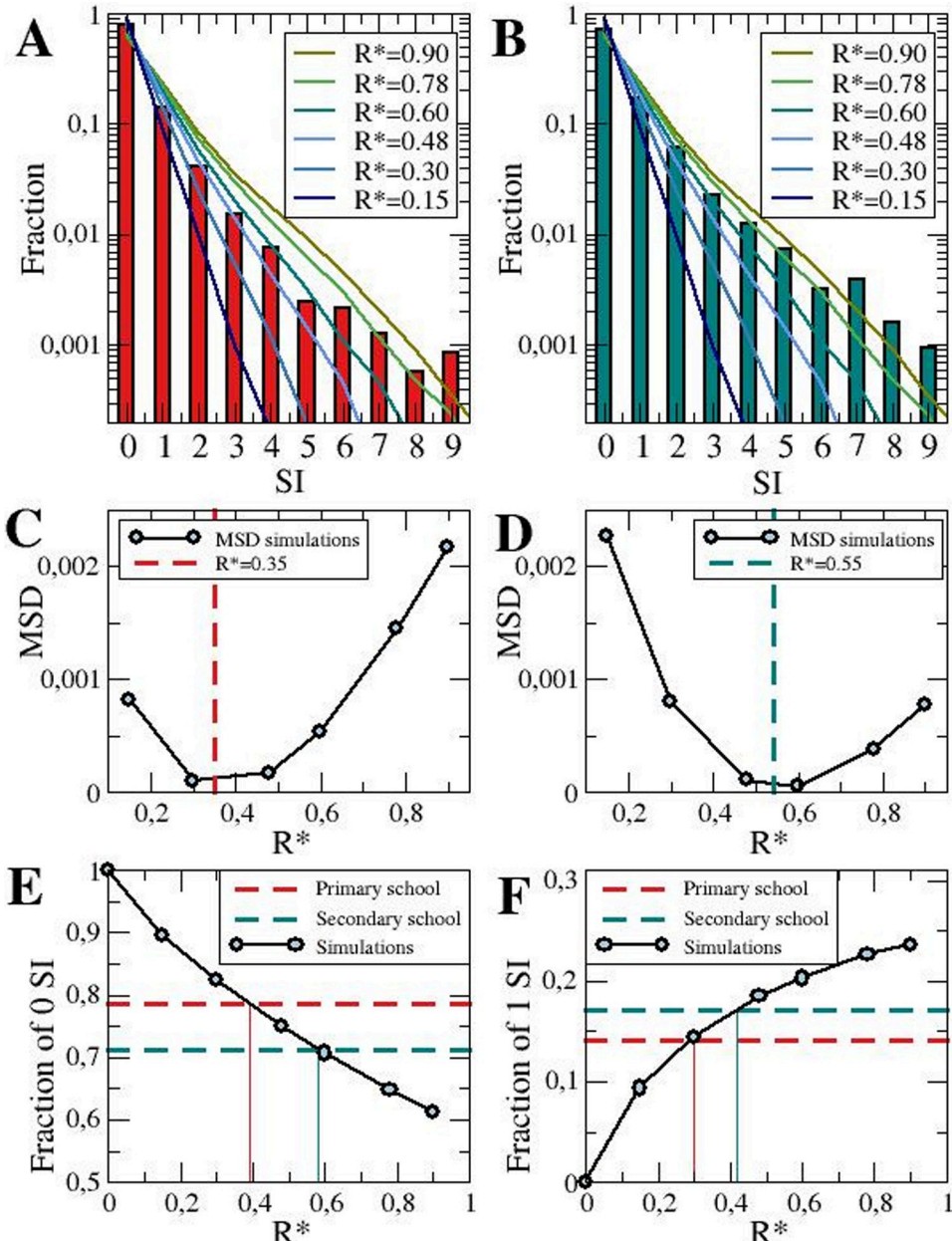

**Fig 4. Estimation of the probability of contagion from epidemiological data.** Fraction of groups going into quarantine with a number N of secondary infected (SI) students by COVID-19 for primary (A) and secondary (B) schools. Bars of the data are shown in comparison with the fraction of groups with a number N of cases of COVID-19 obtained from the numerical simulations (coloured lines) (A and B). Note that y-axis is plotted in a logarithmic scale. Mean square displacement (MSD) of the fractions obtained from the simulations with different values of $R^*$ with the data obtained from Primary (C) and Secondary (D) schools. Vertical dashed lines correspond to the reproductive number calculated from the data obtained from Primary and Secondary schools. Dependence of the fraction of a single case (E) and two cases (F) on the probability of contagion $R^*$ (solid line with circles) in comparison with the fraction obtained in the schools. Red and green dashed lines correspond, respectively, to the fraction in primary and secondary schools of single infected child (E) and two infected students (F). Simulations are done keeping $A_{14} = 250$ cases per $10^5$ inhabitants, and the total number of groups of 20 students fixed to 72000. Data from primary and secondary schools are shown in **S1 Table,** obtained from the Government of Catalonia.

| | 0-9 | 10-19 | 20-29 | 30-39 | 40-49 | 50-59 | 60-69 | 70-79 | 80-89 | 90+ |
|---|---|---|---|---|---|---|---|---|---|---|
| March 2020 | 0,00 | 0,01 | 0,32 | 0,45 | 0,70 | 1,26 | 1,78 | 2,86 | 3,84 | 4,94 |
| April 2020 | 0,01 | 0,02 | 0,43 | 0,43 | 0,56 | 0,84 | 0,83 | 1,36 | 6,33 | 20,96 |
| May 2020 | 0,21 | 0,25 | 0,90 | 0,92 | 0,92 | 1,14 | 0,80 | 0,96 | 2,87 | 8,41 |
| June 2020 | 0,34 | 0,56 | 1,37 | 1,25 | 1,16 | 1,02 | 0,75 | 0,72 | 1,32 | 3,03 |
| July 2020 | 0,56 | 0,80 | 2,72 | 1,99 | 1,38 | 0,93 | 0,38 | 0,29 | 0,41 | 0,78 |
| August 2020 | 0,72 | 0,98 | 2,29 | 1,56 | 1,11 | 0,89 | 0,53 | 0,35 | 0,46 | 0,71 |
| September 2020 | 0,69 | 1,00 | 1,82 | 1,47 | 1,24 | 1,02 | 0,65 | 0,45 | 0,59 | 0,92 |
| October 2020 | 0,40 | 0,93 | 1,54 | 1,31 | 1,30 | 1,21 | 0,86 | 0,63 | 0,84 | 1,39 |
| November 2020 | 0,48 | 1,04 | 1,17 | 1,10 | 1,17 | 1,07 | 0,79 | 0,68 | 1,30 | 2,65 |
| Dec-work 2020 | 0,63 | 1,06 | 1,12 | 1,02 | 0,98 | 0,96 | 0,87 | 0,86 | 1,34 | 2,76 |
| Dec-hol 2020 | 0,63 | 1,15 | 1,30 | 1,03 | 1,06 | 1,03 | 0,99 | 0,92 | 0,80 | 0,87 |

**Fig 5. Heatmap of the ratio between homogenised monthly cumulative incidence by age group and that of Catalunya, excluding data from nursing homes.** Colour scale indicates homogenised incidence below the mean (green) and above the mean (red), as shown by the legend. December has been split between working period (Dec-work) and Christmas holidays (Dec-hol).

### Effect of reopening schools on the epidemiological dynamics

The incidence in the 0–9 years old age group has remained below the Catalan population average systematically despite reopening the schools, during fall 2020 (**S1 Fig**). Nevertheless, the incidence in the 10–19 age range was higher than the average during the September-November 2020 period. This can be partially explained by a higher diagnostic effort in this age group during the last period as part of proactive screening programs. **S2 Fig** shows how the relative number of tests in the 10–19 age range was above double that in the global population during October 2020, and substantially greater than the mean in September (1.3 factor) and November (1.5 factor). This increasing testing was mainly channelled through the secondary schooling centres, where all the classmates of a positive case were systematically screened. Besides, mass testing campaigns in a few institutes within high incidence neighbourhoods were also performed, although this remained uncommon. The relative testing effort in the age range 0–9 was also above the Catalan average during the scholar period. Contrarily, during the Christmas holidays the diagnostic effort in scholar ages was below the Catalan average.

We used the relative homogenized monthly cumulative incidence ($r_{CI_{hom,j}}$) to account for this diagnostic bias and provide a way to compare the epidemiological dynamics among different age groups. **Fig 5** summarizes this ratio ($r_{CI_{hom,j}}$) in each age group with regards to the general population. These ratios are revealing, as they illustrate the most problematic age intervals in each period in a comparable manner. During the first wave, the driving force of the epidemic was established among the elderly ($\geq$70 years of age) although these numbers were heavily biased due to the fact that mild cases were generally not detected, whereas during the months leading to the second wave (mainly summer and early autumn), when testing was already widely available in primary care, contribution to the incidence rested primarily among younger adults (20–59 years old), with the 20–29 age group leading the changes in incidence. Although children played a proportionally greater role during September-December 2020 compared to the beginning of the epidemic, when schools were closed and they were strictly confined at home, new cases occurring in this age group were at a lower or similar incidence than the general incidence in Catalonia.

### Discussion

This set of analyses were designed to document intra-school transmission after school's reopening, to put results in perspective of the general underlying community transmission and to model and evaluate the effect on transmission of different parameters constituting the

prevention measures set up in Catalan schools. It is important however to highlight that the results and conclusions of this study are based on fall 2020 data, and that things may have changed subsequently. During the study period, positive cases detected at schools very closely tracked the underlying incidence rates on the neighbourhoods where those schools were located, and transmission appeared to be adequately contained thanks to the protocols in place in schools, with no evidence of outbreaks, and with individual cases often leading to no further -or very limited-secondary transmission. Indeed ~75% of all index cases detected in schools did not appear to cause any secondary cases, thus limiting the spread of outbreaks, although it is important to highlight that the arrival of asymptomatic infections in children may easily be missed and could trigger undetected secondary transmission. However, transmission chains involving larger numbers of individuals infected will likely trigger the appearance of some symptomatology and thus possibly their detection.

Our study considered propagation inside bubble groups between 15 and 30 individuals. The extension to larger groups like whole schools with more than 500 individuals, may give rise to fictitious higher propagation because the high probability of simultaneous appearance of independent index cases, especially during high incidences outside the schools [35]. In general, the values of R*, smaller than 1, obtained here, were related with low propagation [19] and confirmed that in Catalonia, return to face-to-face activities did not appear to trigger transmission or outbreaks, with schools facilitating epidemiological surveillance [27]. Indeed, during the study period, children did not act as major drivers of the pandemic and in general interventions at schools could be expected to have a small impact on reducing SARS-CoV-2 transmission [30].

Importantly, the relative homogenized monthly cumulative incidence in the paediatric age groups (0–9 and 10–19), although increased during the school months (September-December 2020), appeared to be comparatively much lower than in other age groups. Adults 20–29 played a much more significant role in overall transmission than children. Moreover, incidence rates during the 2020 Christmas break showed a tendency to increase among school-aged children, although these data are confounded by the much lower number of testing during the break. The tracing procedure was not active during this period since the usual school-based protocols were not available. Altogether these data suggested, as other authors had initially modelled [11] or subsequently shown in other countries [17,22,26,36,37], that schools appeared to contribute only slightly to overall community transmission, even in the context of increasing incidence trends characterizing second or third waves, and that their normal functioning, rather than fuelling overall transmission, seemed to have a containment effect, provided that preventive measures were in place and adequately followed. The relatively small number of classes confined, usually proportional to the underlying community incidence at any given moment, together with the virtual absence of any remarkable child-driven outbreak reported, were reassuring evidence that children and schools were not to blame [38], and that the dynamics of the pandemic responded more likely to other driving factors. Older children (10–19 years) seemed to contribute more importantly to transmission that their younger peers, and this suggests that different measures may be necessary to guarantee transmission containing according to age, including for instance the widespread use of COVID-19 vaccines among those aged 12 and older, which were not available during the study period, but which have now been recommended and implemented. However, it is important to highlight that given our results, children of all ages contributed only modestly to community transmission, and as such, they should be allowed to attend physically school, irrespective of age.

We obtained the effective reproduction number inside bubble groups, R*, which it was 0.35 for primary schools (aged from 6 to 11 years) and 0.55 for secondary school (aged from 12 to 17 years) during the study period, which are in agreement with the higher contribution of

older children (10–19) addressed previously. This value is slightly higher than the one obtained by Jordan *et al* [33] in a field study in summer schools, in Catalonia, which was around 0.3. Both results are perfectly compatible, since children attending summer schools were organized in smaller bubble groups of 10 children, and most of the activities were carried outdoors.

Our models also allowed us to test the relative importance of determined parameters which define key characteristics of the preventive strategies in place at schools. For instance, limiting the size of the bubbles appeared only useful when the bubbles were 15 children or less, but transmission seemed not to be substantially increased with larger bubble groups. Similarly, the ideal confinement duration of contacts was 10 days, with transmissibility enhanced with shorter timings, and no additional effect when expanding the duration to two weeks. We did not assess the impact of mask wearing, which in our setting is only compulsory among pupils aged 6 or more, but our data suggest that transmission among the youngest at school remains minor, even in the age group where the use of facemasks is not compulsory according to Spanish recommendations.

One of the big uncertainties regarding this analysis is whether the optimistic scenario observed in Catalonia during school reopening in the school season 2020–21 may change in relation to the introduction of more contagious variants, such as SARS-CoV-2 B1.1.7 (Alpha) or B1.617.2 (Delta), and how these could differentially impact transmission from children at the school level. Although limited, the data that have emerged on the transmission potential of this and other new variants among school children are potentially concerning [24,39], and as a precautionary measure, it would be advisable to closely monitor transmission dynamics in the paediatric age groups and school derived outbreaks in the upcoming months, given the predominance that variant Delta has taken in our setting. Importantly, other countries have used the argument of potential increased transmissibility to close schools but have failed to provide conclusive data on this aspect. Schools in Catalonia have remained fully open throughout the entire 2020–21 academic year and reopened with 100% of the students receiving face-to-face teaching in September 2021, for the new academic year. Transmission trends and school data generated in the next months will again be extremely informative for this purpose. Similarly, this study was conducted prior to the vaccination of teenagers or school children, and further research will be needed to understand the positive additional impact that vaccination may have in intra-school transmission.

Our analyses do suffer from some important limitations worth mentioning. Some of our assumptions have been driven by the little available data in this particular age group, and there is still much to learn about SARS-CoV-2 transmissibility to and from children [40], including the role that the presence of symptoms and the magnitude of the detectable viral loads may play [41]. We also did not differentiate children from adults (which probably have an enhanced transmission potential [42–44]) in our models and did not consider the transmission among bubble groups inside the school. This limits our conclusions in relation to the impact of measures specifically designed for the adult component of school inhabitants, although other authors who have specifically addressed intraschool transmission to adult staff failed to find an associated increased risk [45]. Finally, we were only able to assess transmission in primary and secondary schools, so our conclusions cannot be extrapolated to nursery schools (<3 years of age).

## Conclusions

SARS-CoV-2 infections and COVID-19 cases detected in Catalan schools appear to closely mirror the underlying community transmission but maintaining schools open does not seem to negatively impact community transmission, particularly given that children of all ages

appear to contribute only modestly to onward transmission. Preventive measures in place in those schools appear to be working for the early detection and rapid containment of transmission and should be maintained for adequate and safe functioning of school activities, unless the emergence of new more infectious variants is shown to worsen the observed intra-school transmission patterns. Keeping schools open is allowing children to benefit from face-to-face schooling, a fundamental right that should not be questioned unless stronger evidence emerges proving the contrary. Closing schools should only be considered in the worst-case scenario, where all other possible transmission containment measures have been adequately implemented.

## Supporting information

**S1 Fig. Heat map of the monthly cumulative incidence in Catalonia in cases per $10^5$ inhabitants, from March to December 2020, excluding data from nursing homes.** The heatmap show the incidence in 10-year bin, as well as the global Catalan incidence in the last column. December has been split between working period (Dec-work) and Christmas holidays (Dec-hol).
(DOCX)

**S2 Fig. Heat map of the monthly tests per $10^5$ inhabitants, from March to December 2020, excluding data from nursing homes.** The heatmap show the testing effort in 10-year bins, as well as the global Catalan diagnosis effort in the last column. December has been split between working period (Dec-work) and Christmas holidays (Dec-hol).
(DOCX)

**S1 Table. Distribution of secondary cases per bubble groups in school outbreaks inside primary and secondary schools, and the reproductive number associated for each distribution.**
(DOCX)

**S1 File. Supplementary materials and methods.**
(DOCX)

## Acknowledgments

We acknowledge the group of Department of Education of the Catalan Government involved in the treatment of the data from Traçacovid, for the data shown in S1 Table of this document.

## Author Contributions

**Conceptualization:** Sergio Alonso, Clara Prats, Quique Bassat.

**Data curation:** Bàrbara Baro, Sara Ajanovic, Ermengol Coma, Manuel Medina-Peralta, Francesc Fina.

**Formal analysis:** Sergio Alonso, Martí Català, Daniel López, Enric Álvarez-Lacalle, Clara Prats.

**Investigation:** Iolanda Jordan, Juan José García-García, Carmen Muñoz-Almagro, Eduard Gratacós, Núria Balanza, Rosauro Varo, Pere Millat, Clara Prats, Quique Bassat.

**Methodology:** Sergio Alonso, Iolanda Jordan, Clara Prats, Quique Bassat.

**Project administration:** Joana Claverol, Elisenda Bonet-Carne, Aleix Garcia-Miquel.

**Supervision:** Iolanda Jordan, Juan José García-García, Victoria Fumadó, Carmen Muñoz-Almagro, Rosauro Varo, Pere Millat, Bàrbara Baro, Sara Ajanovic, Sara Arias, Joana Claverol, Mariona Fernández de Sevilla, Clara Prats, Quique Bassat.

**Validation:** Clara Prats.

**Visualization:** Sergio Alonso.

**Writing – original draft:** Sergio Alonso, Clara Prats, Quique Bassat.

**Writing – review & editing:** Martí Català, Daniel López, Enric Álvarez-Lacalle, Iolanda Jordan, Juan José García-García, Victoria Fumadó, Carmen Muñoz-Almagro, Eduard Gratacós, Núria Balanza, Rosauro Varo, Pere Millat, Bàrbara Baro, Sara Ajanovic, Sara Arias, Joana Claverol, Mariona Fernández de Sevilla, Elisenda Bonet-Carne, Aleix Garcia-Miquel, Ermengol Coma, Manuel Medina-Peralta, Francesc Fina, Clara Prats, Quique Bassat.

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
