## [Decision Letter · Decision Letter 0]

19 Aug 2021

PONE-D-21-21706

Individual prevention and containment measures in schools in Catalonia, Spain, and impact of this strategy on community transmission of SARS-CoV-2 after school re-opening

PLOS ONE

Dear Dr. Bassat,

Thank you for submitting your manuscript to PLOS ONE. After careful consideration, we feel that it has merit but does not fully meet PLOS ONE’s publication criteria as it currently stands. Therefore, we invite you to submit a revised version of the manuscript that addresses the points raised during the review process.

ACADEMIC EDITOR: As appended below, the reviewers have raised major concern/critique (reviewer # 2 is against publication) and suggested further justification/work to consolidate the findings. Do go through the comments and amend the MS accordingly. 

We look forward to receiving your revised manuscript.

Kind regards,

A. M. Abd El-Aty

Academic Editor

PLOS ONE

4. Thank you for stating the following in the Funding Section of your manuscript:

“Funding

The components of the analysis drawn from the different KIDS Corona platform have been funded by Stavros Niarchos Foundation (SNF), Banco Santander and other private donors of Kidscorona. We also acknowledge funding from La Caixa Foundation (ID 100010434), under agreement LCF/PR/GN17/50300003; and funding from Ministerio de Ciencia, Innovación y Universidades and FEDER, with the project PGC2018-095456-B-I00. This work has been also partially funded by the European Commission - DG Communications Networks, Content and Technology through the contract LC-01485746. ISGlobal receives support from the Spanish Ministry of Science and Innovation through the “Centro de Excelencia Severo Ochoa 2019-2023” Program (CEX2018-000806-S), and support from the Generalitat de Catalunya through the CERCA Program. CISM is supported by the Government of Mozambique and the Spanish Agency for International Development (AECID). NB is supported by an FPU predoctoral fellowship from the Spanish Ministry of Universities (FPU18/04260). BB is a Beatriu de Pinós postdoctoral fellow granted by the Government of Catalonia’s Secretariat for Universities and Research, and by Marie Sklodowska-Curie Actions COFUND Programme (BP3, 801370).

Role of the funding source

The funders had no role in the interpretation of the data or in the writing up of the manuscript.”

Funding information should not appear in the Funding section or other areas of your manuscript. We will only publish funding information present in the Funding Statement section of the online submission form.

Reviewers' comments:

Reviewer's Responses to Questions

**Comments to the Author**

1. Is the manuscript technically sound, and do the data support the conclusions?

Reviewer #1: Partly

Reviewer #2: No

Reviewer #3: Yes

2. Has the statistical analysis been performed appropriately and rigorously? 

Reviewer #1: N/A

Reviewer #2: No

Reviewer #3: Yes

3. Have the authors made all data underlying the findings in their manuscript fully available?

Reviewer #1: Yes

Reviewer #2: No

Reviewer #3: Yes

4. Is the manuscript presented in an intelligible fashion and written in standard English?

Reviewer #1: Yes

Reviewer #2: No

Reviewer #3: Yes

5. Review Comments to the Author

Reviewer #1: The authors tackle an important health issue in whether schools should stay open or close due to COVID-19. They have access to a unique dataset that purports the number of secondary infections in schools in Catalonia. They then constructed models to estimate the transmission within schools and how school aged children contributed to the overall transmission of COVID-19 in Catalonia.

I have listed the comments in chronological order of the manuscript.

Lines 283-3 pg 11. The authors mention that the global detection rate of COVID-19 in children is 70%. How did the authors choose this number? Is it based on the literature? If so, please cite?

Lines 286-7 pg 12. Can the authors explain why they chose to keep community transmission at R = 1? At such a low R would there even be a debate about whether or not to close schools? Given the access to data the authors had, couldn’t they have estimated R in the community during the time of their simulation and made R variable based on estimated actual conditions?

Lines 305 pg 12. The authors estimate that children spend 30% of their time in school and thus the R in school is 30% of the total R. Can the authors clarify how they calculated 30%? First off, does this fraction take into account weekends? Also, have the authors taken into account that ~33% of the day that children are asleep and are highly unlikely to infect anyone if they sleep alone and can probably only infect whoever they share a room with? Would the fraction make more sense based on the number of waking hours they spend in school?

Lines 339-341 pg 13. The authors state that they evaluated four scenarios but list five different size bubble groups. Is this a typo or am I missing what the four scenarios are?

Lines 348-9 pg 14. The authors state that they are evaluating R* from 0.3 - 1.5, which translates to an R of 1 – 5. Can the authors comment on why they chose to study the range of R values they did? Are these in line with the range in R that was calculated in Catalonia during the study period? Also, it may be helpful to include both R* and R in the figure legends to give the numbers more context.

Figures 1 and 2. The authors state that the model is stochastic and that they have multiple runs, yet the plot is represented as a bar plot. Did all runs produce the exact same number of secondary infections? If not, can the authors please replot and replace the bar plots with boxplots to show how much variation there is in the model?

Figures 1 and 2. While the number of secondary infections is interesting, it might be more useful to show the number of secondary infections per 10,000 students in order to give a larger context on the severity of the spread of COVID-19. While the authors do provide the number of students in the figure caption plotting rates on the y-axis would make it easier to evaluate the impact of secondary infections according to the model.

Lines 365-371 pg 14. The authors have great data on the impact of bubble groups in Catalonia. Do the authors by any chance have the distribution in sizes of the bubble groups (i.e., what percentage were of size 20)? Also, how did the authors determine that the size of the bubble groups is most likely 20?

Line 375 pg 14. The authors state “a statistically significant difference…” Unlike many manuscripts this manuscript does not rely heavily on results driven by p-values and I commend the authors for this effort. I believe in this instance that the authors are referring to the results of another manuscript, but would appreciate if they removed “statistically significant” from this manuscript. Perhaps they could simply delete “statistically significant” and leave “a difference…” The reason for this comment is that the statistics community has recently come out strongly against p < 0.05 and away from using the terms statistical significance, which to their community is not a term that has meaning (please see Wasserstein et al. 2019 https://doi.org/10.1080/00031305.2019.1583913 ; also for a brief history of p-values see Nuzzo 2014 - https://www.nature.com/articles/506150a). I understand that such a transition will take time but would encourage the authors to change their language in order to help promote better science. Thank you in advance.

Line 385 pg 15. The authors talk about incidence accumulated over 10 days (A10). Can the authors please clarify if this the cumulative incidence reported on the date in question or is it the actual incidence for the 10 days prior to the date in question? i.e., could this model be used to predict based on the current known data? Or is calibrated to modeling weeks or months after the fact?

Lines 389-395 pg 15. The authors mention uncertainty in the effective time spent in school and the probability of detection. Have the authors tried using their model to search this parameter space to get a better sense of the most likely percent of effective times spent at school and the percent probability of detection? Getting a sense of the range and likelihood of these values would be an important contribution to informing whether schools should open.

Line 407 pg 15. There is an empty box followed by a * instead of a symbol. I imagine this was some sort of file conversion error and wanted to give you a heads up.

Lines 427-432 pg 16 – Supplementary Table 1. Can the authors please clarify the cases column in supplementary table 1. It looks as if that for every primary infection there is at least one secondary infection. Perhaps shift the scale to start at 0 and rename secondary infections.

Figures 4 A and B. Again, I believe that “Number of Infected Kids” is meant to show the number of secondary infections where 1 means there were no secondary infections. I believe it would be easier to understand these plots if the scale was shifted and the x-axis was renamed Number of Secondary Infections. Also, since the y-axis is log-scale and the first column is near 0.7 or 0.9 it might be helpful to include minor gridlines to better see the difference, which looks small on log scale but is quite large in terms of this study.

Lines 437-8 pg 16. Can the authors comment on how the results of the model would change if the R outside of schools > 1? Wouldn't this be worth exploring since often the biggest driver of people wanting to shut down schools is when R > 1 in the community.

Lines 438-9 pg 16. The authors state that the reason they did not run R* = 0 but did not state why they choose R* = 0.15 – 0.9. Can the authors comment on why they chose to model this range of R* values?

Lines 440-2 pg 16. The authors state that their model fits the data from the schools. Can the authors state what are the predicted R*s for each school? How did the authors determine which R* was the better fit (i.e., what measurement did you calculate to determine the best fit)?

Figures 4C and 4D. Please add figure legends.

Line 460 pg 17. Small typo. The authors wrote “stablished” when I believe they meant to write “established.”

Figure 5. Please add a colorbar.

Lines 517-20 pg 19. I love the point the authors make regarding schools only having a slight contribution to overall transmission. That being said the results only show schools having a slight contribution with the variants of COVID-19 studied. New variants such as B1.617.2 are more transmissible. Have the authors looked into incorporating a more transmissible variant into their model and how much schools could contribute with said variant? Also, can the authors quantify how much schools contribute to overall transmission (e.g., 10% to overall transmission or 20% less than expected)?

Lines 547-8 pg 20. The authors state “but our data suggest that transmission among the youngest remains minor, even in the absence of facemasks.” How do the authors justify this statement? From what they state the models are based on school data where students were required to wear masks. Please justify or remove.

Line 551 pg 20. Why do the authors mention B1.1.7 but do not mention the more prevalent and more transmissible B1.617.2? Also, these variant names may be a bit too jargony for this manuscript. Perhaps replace or add alpha and/or delta.

Lines 581-3 pg 21. The authors state “Closing schools should only be considered in the worst-case scenario, where all other possible transmission containment measures have been adequately implemented.” Based on this study could the authors state what a worst-case scenario would look like. Also, can this model be used to advise schools when they are approaching a worst-case scenario?

The authors show that having schools open did not greatly impact the greater community and also did not lead to many known secondary infections in school, which I find to be one of the most interesting findings in this manuscript. However, we originally shut down schools during this pandemic because we looked at how important schools were in the transmission of diseases during previous pandemics and epidemics. Therefore, we should be cautious in implying that because past school closures due to COVID-19 were not necessary to limit transmission because that does not necessarily mean that future schools closures due to COVID-19 are not necessary given the increased transmissibility of new variants.

Reviewer #2: #1. The study presents the results of original research.

YES

The manuscript is devoted to the analysis of the 2020 Autumnm term of the academic year in Catalonia Schools using a public database of infections in Schools.

Although there is no clear goal or main outcome of the paper, the main points of the manuscript are the following:

(G1) The authors do have an opinion on the importance of presential school, which I think would be more suitable for an opinion letter to a scientific journal.

(G2) The authors describe some aspects on the number of students infected, classes confined and number of secondary cases per infected student.

(G3) The authors present a 'modeling exercise' (in their own words) to "demonstrate the limited role of childhood transmission in relation to overall community transmission when proper epidemiological surveillance and recommendations are in place." To this end, their intention is to compare the output of their simulations with data from schools and incidence in Catalonia during the same period.

A web search of "SARS School Catalonia" reveals several other papers on this dataset that have not been commented on the present manuscript. The authors should compare their findings with existing analysis of the same dataset.

#2. Results reported have not been published elsewhere.

NO.

The description of infections in classrooms (item (G2) above), for instance what they call the reproduction number or the number of primary cases which define no secondary infections, have been partly reported by some of the authors in the paper:

"Age-dependency of the Propagation Rate of Coronavirus Disease 2019 Inside School Bubble Groups in Catalonia, Spain". doi: 10.1097/INF.0000000000003279

#3. Experiments, statistics, and other analyses are performed to a high technical standard and are described in sufficient detail.

NO

The manuscript compares data from the schools registry (item (G2) above) with the outcome of their simulations (item (G3) above).

There is no statistical framework which supports the comparison of data with the result of the simulation. The authors use the word "significant" without referring to a statistically significant test, inducing confusion.

The simulations are a not presented in a standard fashion and their desription is insufficent, in my opinion. Many important parameters like Rmax or the probability distribution that a positive child is detected, for instance, are not properly described. In my opinion, the simulation cannot be reproduced. Given the many unknowns around the infection processs parameters, any simulation should include estimates of uncertainty or at least a thorough sensitivity analysis. Lack of code availabaility is also problematic given the computational approach.

#4. Conclusions are presented in an appropriate fashion and are supported by the data.

NO

A major issue with this manuscript is the inclusion of several conclusions which have important implications either clinically or for policiy makers and are not supported by the data nor the analysis.

(1) L 541-544: The authors state that "transmission seemed not to be significantly increased with larger bubble groups". There is no statistical test which could yield significance and the conclusion on the size of groups, which has strong policy implications, is not supported by results.

(2) L 544-546: "the ideal confinement duration of contacts was 10 days, with transmissibility enhanced with shorter timings, and no additional effect when expanding the duration to two weeks." or "Our models suggest that the optimal confinement time for the contact of a positive case is 10 days" (L 141-142). In fact, their assumption is that "the detected kid always stays at home time enough to recover at home and become non-infectious" (L291) which further limits their analysis.

(3) L 546-548: "We did not assess the impact of mask wearing, which in our setting is only compulsory among pupils aged 6 or more, but our data suggest that transmission among the youngest remains minor, even in the absence of facemasks". This is not even discussed in the paper and the conclusion on the lack of effect of face masks contradicts recent CDC guidance that masks should be worn indoors by all individuals (age 2 and older). https://www.cdc.gov/coronavirus/2019-ncov/community/schools-childcare/k-12-guidance.html

(4) In "What do these findings mean?" it is stated that "School reopening (...) is safe for children and adult staff" although the only reference to adults in the paper is in the conclusions (lines 566-570) when they acknowledge the impossibility of deriving conclusions on the "adult component of the school inhabitants".

(5) The authors state in the abstract that their results imply "decreased transmissibility for children in comparison to adults". I think this is not supported by the analysis presented.

The findings of the article are difficult to put in context because the authors use some non-standard terminology and definitions of important notions are lacking. We mention the following:

(1) OUTBREAK. There is no definition of "outbreak" (index case, follow-up period, etc...). Besides, there are contradictory statments on this concept:

- Abstract: "(...) scarce evidence of significant intraschool transmission in the form of outbreaks." (also in the methods section).

- lines 523-524 virtual absence of any remarkable child-driven outbreak reported (this conclusion is given without justification).

- Table S1 "Distribution of secondary cases per bubble groups in school outbreaks inside primary and secondary schools". This implies hundreds of outbreaks.

(2) PRIMARY vs SECONDARY CASES: How are cases classified between primary and secondary?

(3) R and R*: One of the main outcomes of the manuscript is the "effective reproductive number" but no definition is given. Although the effective reproduction number may seem a simple concept its definition is tricky and has to take into account the epidemiology of the particular disease. The definition of R is referred to a preprint by the some of the authors (which is not included in the references). Although the authors refer to the work of Li et al [24] on household transmission they do not use a similar approach. Besides, it is not correct to compare several values of the reproduction number (in "the community" or "schools) obtained from several definitions and to draw conclusions about them. The authors do not seem to use any standard technique to assess the value of R in the general population.

(4) The authors define an indicator termed "homogenized cumulative incidence" (L252) which is a modification of the incidence. If the authors wish to use the indicator they define and draw conclusions, they need to properly analyze this quantity. This methodology would require an statistical analysis which is absent in this manuscript nor in the references they use.

(5) Tests. Did the protocol use PCR-testing or other rapid tests? There is a contradition also with mass screenings. L237: "Although originally planned, no mass screening and testing of teachers or pupils was conducted during the first trimester" but later in line 474: "Besides, several mass testing campaigns in the institutes within high incidence neighbourhoods were performed".

#5. The article is presented in an intelligible fashion and is written in standard English.

NO

Althought the article is written in a correct standard English, the presentation suffers from several important problems. Apart from the absence of definition of the key objects presented in item #5 above, the structure and writing missess important aspects:

- Study period? There is no study period defined. I understand that it is more or less the Autumm term, but precise dates are missing and conflicting dates are given in different places of the paper. This happens, for instance, in lines 243, 275, 286 and 760.

- There is no clear IMRaD structure: results and conclusions are scattered all over the paper.

- The manuscript needs to be updated (for instance, refers to B.1.1.7 as a potential threat). Updated systematic reviews should be included in the introduction. Some sentences in the introduction are stated without justification.

#6. The research meets all applicable standards for the ethics of experimentation and research integrity.

NO

The ethics statment is confusing: it refers to "summer camps" and informed consent, but it is not clear which students or adults had an informed consent because no individual data from summer camps is used.

7. The article adheres to appropriate reporting guidelines and community standards for data availability.

NO

Although some data appears to be restricted for data protection, no adequate summary is presented (no table 1 with demographics description). Lack of code availability and diagnostic tests for the simulations limit the intepretation of the simulation.

Reviewer #3: This is an interesting manuscript aiming at studying the impact of the reopening of schools in the SARS-CoV-2 community transmission in Catalonia, Spain. Focusing on a children population is smart as it has not been well described in the literature and because school closure may have consequences on the kids' education and mental health. Results of the study showed that SARS-CoV-2 infections in schools mirror the community transmission. Closing schools should be the last resort option to limit the spread of SARS-CoV-2 when all other preventive measures have already been implemented.

Minor issues :

1. It would be interesting to discuss more about the vaccination against COVID-19 as children between 12-18 years old are eligible for being vaccinated and how it might impact the R* and therefore the transmission in schools and in the community.

2. It is known that symptomatic persons are more likely to be able to spread the virus compared to asymptomatic persons. Is there a difference according to the presence of symptoms in terms of R* among children in schools ?

3. Rounding decimals would improve the legibility of the Suppl Table 1.

4. Suppl Table 1 appears twice in the document

6. PLOS authors have the option to publish the peer review history of their article (what does this mean?). If published, this will include your full peer review and any attached files.

Reviewer #1: **Yes: **John P. Hanley

Reviewer #2: No

Reviewer #3: No

---

## [Author Response · Author response to Decision Letter 0]

4 Oct 2021

We thank the editors ad reviewers for the thorough revision, and we have tried to address all comments in a pint-by-point response letter. Given the length of the responses (document of 8 pages), we have opted for attaching it as a document in the portal.

---

## [Decision Letter · Decision Letter 1]

1 Nov 2021

PONE-D-21-21706R1Individual prevention and containment measures in schools in Catalonia, Spain, and impact of this strategy on community transmission of SARS-CoV-2 after school re-openingPLOS ONE

Dear Dr. Bassat,

Thank you for submitting your manuscript to PLOS ONE. After careful consideration, we feel that it has merit but does not fully meet PLOS ONE’s publication criteria as it currently stands. Therefore, we invite you to submit a revised version of the manuscript that addresses the points raised during the review process.

ACADEMIC EDITOR: Still, reviewers are raising substantial concerns over the revised form of the MS (reviewer # 2 is still against publication). Would you please go through their comments amend the MS accordingly? Afterwards, proofread the MS for grammar and syntax errors.

We look forward to receiving your revised manuscript.

Kind regards,

A. M. Abd El-Aty

Academic Editor

PLOS ONE

Reviewers' comments:

Reviewer's Responses to Questions

**Comments to the Author**

1. If the authors have adequately addressed your comments raised in a previous round of review and you feel that this manuscript is now acceptable for publication, you may indicate that here to bypass the “Comments to the Author” section, enter your conflict of interest statement in the “Confidential to Editor” section, and submit your "Accept" recommendation.

Reviewer #1: All comments have been addressed

Reviewer #2: (No Response)

2. Is the manuscript technically sound, and do the data support the conclusions?

Reviewer #1: Yes

Reviewer #2: No

3. Has the statistical analysis been performed appropriately and rigorously? 

Reviewer #1: N/A

Reviewer #2: No

4. Have the authors made all data underlying the findings in their manuscript fully available?

Reviewer #1: Yes

Reviewer #2: No

5. Is the manuscript presented in an intelligible fashion and written in standard English?

Reviewer #1: No

Reviewer #2: Yes

6. Review Comments to the Author

Reviewer #1: I appreciate the time and care the authors took in responding to the reviewer comments. I also empathize with the long review process the authors have experienced resulting in the delay of the dissemination of this work. The manuscript is much improved, but there are still a few minor comments to address.

Below are some general themes that I feel should be addressed.

Overall, a lot of work needs to go into the improvement of word choices and grammar (e.g., lines 178-9 “children were subject to the most severe of the lockdowns.”). Unfortunately, PLOS ONE does not copyedit, so I would suggest seeking help with the wording and grammar if necessary.

Another general theme is putting this work in temporal context. Often the authors use present tense to describe the transmission of COVID-19 in schools. While I acknowledge that the present tense makes for better scientific writing, I am afraid that given the dynamic nature of this pandemic that readers may be under the impression that the results in this manuscript can be directly translated into the present reality of the pandemic which now has new variants and vaccines. I know the authors mention the variants and vaccines at the end of the manuscript, but I feel that an earlier emphasis on the fact that the results and conclusions of this study are based on Fall 2020 data would help prevent any confusion.

I list more specific comments below in chronological order below.

In the conclusion of the abstract the authors state that “…cases detected in Catalan schools appear to closely mirror the underlying transmission from neighborhoods…”. This is a small, but I believe to be an important comment which is that more emphasis needs to be placed on the fact that this data and consequently the conclusions are tied to the COVID-19 pandemic in Fall 2020. I think it is important to clarify that any “mirroring” of underlying transmission and the success of preventative measures are all based on the pandemic pre-delta variant and pre-vaccinations. If the authors could add this disclaimer to the conclusion of the abstract I think it will help eliminate any confusion as to whether the same dynamics are driving the pandemic in the Fall of 2021. I would take the same approach to lines 147-149 under what did the researchers find? And under the section what do these findings mean? Lines 151-155 should also be qualified to acknowledge that the results in this manuscript and subsequent conclusions only apply to the COVID-19 pandemic pre-delta variant and pre-vaccines.

Line 168 typo? I think the authors mean to write “containment” instead of “contention”

Line 208: Please be consistent in your word choices. On line 208 you use “pre-school centres” which I believe you defined as “pre-primary schools” on lines 205-6. I apologize if these two school types are different, but if they are different then please define “pre-school centres” alongside pre-primary schools.

Lines 367-8: In Figure 1’s caption: “Nevertheless, the role of the size of the bubble group seems relevant only in scenarios of high transmission rate.” I do not see any data or plots related to bubble group size in this figure. This may be due to mixing up the order of the Figures when they were uploaded or mixing up the order of the figure captions since Figure caption 2 seems to correspond to Figure 1 and vice-versa.

Figures 4 & 5 appear to be missing captions in the body of the text making it hard to interpret the plots. I found the captions later in the manuscript and now see that the authors used MSD to determine the “best” R* for primary and secondary schools. Based on the curves, I think it would be worth commenting in the manuscript that the “best” R* is good at predicting a low number of secondary infections but are very poor at estimating a higher number of secondary infections (i.e., at 9 secondary infections an R* closer to 0.9 is better. Overall Figure 4 looks to busy with axis labels blending into different subplots, please clean this up.

Lines 531-2: The authors state that “schools were a key containment measure tool.” This statement seems to suggest that if children did not go to school then transmission in the community would be higher. I am not sure that the work in this manuscript shows that to be the case. I do however agree with the statement that schools “facilitated epidemiological surveillance.” Later in the paragraph the authors continue to state that schools contain the virus, which I do not believe is proven in this manuscript. However, the authors are correct to state that R* is likely to be well below 1 and thus schools are not likely to result in outbreaks. Again, all this analysis needs to be in the context of pre-delta variant since schools in the US did have outbreaks amongst the students when the delta variant was present in Fall 2021. The authors do mention at the end of the discussion the context of alpha and delta variants as well as vaccinations. However, I feel it would be a stronger manuscript if the context of when this study was conducted receives more emphasis. One solution would be to change verb tenses from the present to the past when discussing the results since this will imply to the reader that the findings in this manuscript may no longer apply to the current time in the pandemic.

Overall, I find this to be an important piece of work that will add to the discussion of whether or not to close schools during this pandemic and future pandemics. This manuscript shows that having in-person education may not result in large outbreaks for some diseases, such as COVID-19 pre-variants. These findings will serve as important information for any future scientific and policy discussions.

Reviewer #2: The authors have made some changes in the writing of the paper and corrected some of the errors and inconsistencies pointed by the different referees. For instance, they have published the code of the simulation in a public repository. Some other questions, like the definition of outbreak throughout the paper or the full description of the dataset have not been resolved. The article needs to be compared with current evidence on COVID-19 and school opening. The authors still describe their analysis as "modelling exercises" (L347).

In my previous report, I recommended rejection because, in my opinion, the conclusions of the paper are not supported by the data or the analysis. There is no statistical analysis which can measure the association or causality of school opening on incidence. In order to present conclusions like "SARS-CoV-2 infections and COVID-19 cases detected in Catalan schools appear to closely mirror the underlying community transmission from the neighbourhoods where they are set" the authors need to present a framework in which the question of association can be considered.

7. PLOS authors have the option to publish the peer review history of their article (what does this mean?). If published, this will include your full peer review and any attached files.

Reviewer #1: **Yes: **John P. Hanley

Reviewer #2: No

---

## [Author Response · Author response to Decision Letter 1]

5 Nov 2021

We thank the reviewers for the thorough revision, and we have tried to address all comments in a point-by-point response letter.

---

## [Decision Letter · Decision Letter 2]

6 Dec 2021

PONE-D-21-21706R2Individual prevention and containment measures in schools in Catalonia, Spain, and impact of this strategy on community transmission of SARS-CoV-2 after school re-openingPLOS ONE

Dear Dr. Bassat,

Thank you for submitting your manuscript to PLOS ONE. After careful consideration, we feel that it has merit but does not fully meet PLOS ONE’s publication criteria as it currently stands. Therefore, we invite you to submit a revised version of the manuscript that addresses the points raised during the review process.

ACADEMIC EDITOR: Unfortunately, both reviewers of the previous revision (R1) declined to assess the revised MS. Therefore, I’ve invited fresh reviewers to evaluate the MS. As you can see, they still raising some concerns over the content. Would you please go through the comments and amend the MS accordingly? 

We look forward to receiving your revised manuscript.

Kind regards,

A. M. Abd El-Aty

Academic Editor

PLOS ONE

Reviewers' comments:

Reviewer's Responses to Questions

**Comments to the Author**

1. If the authors have adequately addressed your comments raised in a previous round of review and you feel that this manuscript is now acceptable for publication, you may indicate that here to bypass the “Comments to the Author” section, enter your conflict of interest statement in the “Confidential to Editor” section, and submit your "Accept" recommendation.

Reviewer #4: (No Response)

Reviewer #5: (No Response)

Reviewer #6: (No Response)

2. Is the manuscript technically sound, and do the data support the conclusions?

Reviewer #4: Partly

Reviewer #5: Partly

Reviewer #6: No

3. Has the statistical analysis been performed appropriately and rigorously? 

Reviewer #4: Yes

Reviewer #5: I Don't Know

Reviewer #6: No

4. Have the authors made all data underlying the findings in their manuscript fully available?

Reviewer #4: No

Reviewer #5: Yes

Reviewer #6: Yes

5. Is the manuscript presented in an intelligible fashion and written in standard English?

Reviewer #4: Yes

Reviewer #5: No

Reviewer #6: No

6. Review Comments to the Author

Reviewer #4: Since I join the review process from the current version, I do not intend to delay it. Thus all my comments are suggestive, not mandatory; I just want to improve the manuscript.

I do not understand why the paper examines a wide range of R* values. It seems that the authors only have to focus on the estimated values, 0.35 and 0.55. Otherwise, I am curious why they choose the specific values of 0.3, 0.6, …, and 1.5 (p. 14).

I believe there remains much room to improve the way the paper presents in its figures. For instance, Figure 1 displays bar graphs where my eyes focus on differences among R* values, though the authors seem to call attention to variation between bubble size (p. 12). An alternative is line graphs. Similar comments hold for Figure 2 as well.

I am afraid some notations lack explanation, so that it is not easy to read.

- On p. 11, I do not understand why the paper calculates “the homogenized cumulative incidence” as they do. Why does it multiply the cumulative incidence by the positivity rate? And then why does it divide it by a baseline? Why is the baseline 5%?

- On p. 12, R^X and N_index suddenly appear without explanation. I can imagine what they mean, though I would like to see their definition.

- In Figure 4, what does “SI” stand for? Secondary infection or single infected kids? Accordingly, I do not understand Figures 4A and 4B. By the way, the range of R* values considered is 0.15 to 0.9, which is different from those of Figures 1 and 2. Why?

- In particular, I recommend not using the word “bubble” (without definition) in the abstract. My sense is that it is not such a common usage.

Reviewer #5: Major Comments:

“Individual prevention and containment measures…” is a major article that uses mathematical modeling informed by real-world data to estimate 1) the R* in school settings following initial re-opening during the 2020-21 school year in Catalonia, and 2) to estimate the impact of two specific strategies (quarantine duration and bubble size) on in-school transmission. Similar to other studies, the authors find that the R* in the school setting was low (although this finding was previously published by some of the same authors in Pediatric Infectious Diseases Journal: https://journals.lww.com/pidj/Fulltext/2021/11000/Age_dependency_of_the_Propagation_Rate_of.2.aspx). Despite the study’s title, the authors did not really measure the impact of school opening on community transmission, and any mention of the impact of school opening on community transmission should be removed from the title, abstract, etc, and moved to the discussion/background sections. Other recently published manuscripts have addressed this question (e.g., Ertem et al, Nature Medicine: https://www.nature.com/articles/s41591-021-01563-8).

Throughout the manuscript, methods and results are intermixed. These sections need to be completely re-organized and re-written, such that methods are clear and that the results follow from the methods described. In addition, the authors need to be clear about what they did and did not study – specifically, they modeled quarantine and bubble size as infection control strategies. Those should be the focus of all aspects of the manuscript – the rest does not follow from their results.

Specific Comments:

Abstract:

Methods/results not clear and are not organized. Please restructure.

Conclusions: Please remove any discussion of the impact of schools on community transmission. That was not studied in this manuscript (in addition, a simple correlation is not sufficient to evaluate this question, as the impact of a policy change is not seen in inicidence rates for 3-6 weeks. Community incidence rates reflect new introductions into schools, and then a lag period is needed to see the impact of school opening on community transmission- this would not be expected to occur instantaneously, but rather with a lag which is not addressed in the abstract or the manuscript).

Author Summary

-Clarify if surveillance was mandatory

-In the second bullet, clarify that the mathematical model used real-world inputs

-In the final bullet, add “pre-vaccine” in addition to “pre-delta”

-In the what do these findings mean? Section, remove sections on impact of school opening on community transmission (first bullet) and the bullet on closing schools as a last resort (last bullet)– their study provides no insight into this question

Manuscript

-Lines 170-171: Rephrase to “the extent to which children of varying ages can be effective drievrs of supershedding events” for clarity

-Lines 175-179 – It is not clear that schools closed pre-maturely – early in the pandemic, very little was known. The decision for school closures was based on models of influenza, which may have been appropriate early, even if we learned otherwise later.

-Lines 190-191 – add recent publications about impact of school mode on community transmission. There are several papers published on this topic (See Ertem et al).

Methods/Results

-See major comment above. These sections need to be entirely reorganized and re-written such that there is a clear methods section and a clear results section. Currently, these are entirely intermingled.

-The entire section is highly technical and difficult to understand for those without a statistical background. Strongly recommend re-writing such that this is more understandable for a more general scientific audience.

-Lines 495 – 525 – Authors appear to have used a simple correlation to evaluate impact of schools on community rates, however, this method is fundamentally not a valid analysis. Schools exist as part of a community infrastructure, such that a case from the community is introduced into a school, where it can spread, and then that second case can spread to others in the community, which can then lead to more introductions into the school setting and the potential for more spread. However, this process takes time. There is a lag between a primary and a secondary case, such that a week to week correlation is not the appropriate measure for identifying the impact of school opening on community spread – a window period needs to be considered. This section could be rephrased to simply state that cases in schools tended to be lower than cases in the community at the same point in time, and any discussion of the impact of schools on community rates removed, as the authors did not evaluate this question.

Discussion

-As noted repeatedly above, the authors did not study the impact of schools on community transmission, but rather modeled transmission in schools (which the authors previously published) and the impact of specific infection control measures in school settings on in-school transmission. This should be the focus of all of their statements about what they did and did not find.

-While it is accurate that the study was conducted pre-delta, it was also conducted pre-vaccine, and both developments are likely to impact the findings in unpredictable ways. Both should be discussed/acknowledged.

Reviewer #6: Review of “Individual prevention and containment measures in schools in Catalonia, Spain, and

impact of this strategy on community transmission of SARS-CoV-2 after school reopening”

PLOS One.

I think this paper has some valuable material and provides interesting results about school transmission, but there are some problems that make it still fairly far from acceptable for PLOS One at this time.

A lot of it is just very unclear. For example, CI_hom is introduced and I could not figure out what the motivation was. Why not just use incidence? Reading further, I gather that they want to factor in imperfect ascertainment. This was not mentioned earlier, and even assuming that I don’t understand the mystery factor of 0.05. There are lots of things like this.

There is some tone that doesn’t belong in a scientific paper in my opinion. I agree with the authors that school closures for COVID-19 were not the best policy (with the benefit of hindsight) but there are a few spots which I detail below where they take this for granted, whereas the job of the paper is to persuade the reader of this.

From reading the abstract I thought they were going to just estimate the number of transmissions from each index case by counting the number of cases in a cluster and subtracting one. Maybe factoring in imperfect ascertainment would be good, but the surveillance seems pretty good. However, as I got into the paper there is a model that is much more complicated and the R* is estimated by tweaking parameters in the model and fitting to data. This is fine, but I would like this to be explained better.

Page 4, I didn’t understand the sentence on lines 93 to 97

“the proportion of “bubble groups” (stable groups of children doing activities together) confined ranged” So only 1 to 5% of all students were in a bubble group? Or bubble groups each had between 1 and 5 students? And I don’t see the connection with the other parts of the sentence. I think “confined ranged” is maybe a typo.

Line 98 “the effect of different parameters part of the defined preventive strategies” I couldn’t parse this.

Line 104 “Homogenized monthly cumulative incidence (CIhom) was assessed to compare the epidemiological” I don’t know what homogenized monthly cumulative incidence is. Can you rephrase in some way to make it clearer what you’re doing? Also, what is the subject of the verb “suggested”? The CI_hom? Does it suggest something? Or does your analysis suggest it?

Page 6, line 124. I think the phrase “extraordinarily infrequent” is too strong. (My numbers are around 1 in 200 children with COVID infection are hospitalized.

Page 8, line 175. The word “premature” contains a value judgement that has not been justified yet. You could say it was used prematurely.

Line 189, “re-state” Not the right word. “Reignite’?

Lines 191 to 192. “but again, in the context of insufficient evidence of the real contribution of children and schools to ongoing transmission” I find the tone here inappropriate. This is a scientific paper, not a blog post or an editorial page. The authors’ job (if they want to persuade the reader) is to provide evidence that schools are safe and that closures are a bad idea. Saying that there was insufficient evidence to close schools with a new variant is a value judgement. I don’t think it belongs in the introduction. You can summarize a situation and the decisions made without broadcasting your judgment in every sentence.

Page 9, line 196-197. This is not a sentence, but a fragment. (No predicate.) Same with the following sentence.

Page 10, line 225. Suggest replacing “agile” with “rapid”.

Page 10-11. I do not understand the purpose of “homogenized cumulative incidence”. Firstly, would cumulative incidence just be the total number of detected cases in a time interval? Secondly, what is the positivity rate? The rate at which PCR tests come back positive? Why would I multiply that by incidence? And divide by 0.05?

Page 11 “Although it does not aim to evaluate the real incidence” Why is it better than CI_j?

Page 12, line 291 “kid” is informal in English. Use “child” or “student”

Line 315, do you mean the superscript to be x in this equation?

Page 13, The section heading say “Statistical Methods” it starts with talking about bubbles. I don’t see any statistical method discussed here.

Page 15. I’m somewhat skeptical of the good fit in Figure 3, especially given that there were no free parameters. Or did the authors choose the parameters like 0.35 to get a good fit? If they did, that would be fine, but they should be transparent about it. I agree that the number of cases in schools appears to track the number in the community very closely.

Page 18. I didn’t find this to be a compelling analysis. I actually agree that school reopenings were unlikely to have a huge effect on community transmission, but this is not systematic investigation. What are they estimating exactly? I think the thing that is persuasive is just that there is apparently not much transmission in schools. You can’t infer much from one single reopening event in one jurisdiction where you observe what happens to incidence.

Page 22, “presential” is not a commonly used English word. How about “present” or “current”.

”Keeping schools open is allowing children to benefit from presential schooling, a fundamental right that should not be questioned unless stronger evidence emerges proving the contrary” Again, I agree, but stating that something is a right is an ethical statement, not a scientific one.

7. PLOS authors have the option to publish the peer review history of their article (what does this mean?). If published, this will include your full peer review and any attached files.

Reviewer #4: No

Reviewer #5: No

Reviewer #6: No

---

## [Author Response · Author response to Decision Letter 2]

22 Dec 2021

We thank the editors and reviewers for the thorough revision, and we have tried to address all comments in a point-by-point response letter (See attached)

---

## [Decision Letter · Decision Letter 3]

26 Jan 2022

Individual prevention and containment measures in schools in Catalonia, Spain, and community transmission of SARS-CoV-2 after school re-opening

PONE-D-21-21706R3

Dear Dr. Bassat,

We’re pleased to inform you that your manuscript has been judged scientifically suitable for publication and will be formally accepted for publication once it meets all outstanding technical requirements.

Kind regards,

A. M. Abd El-Aty

Academic Editor

PLOS ONE

Additional Editor Comments (optional):

Reviewers' comments:

Reviewer's Responses to Questions

**Comments to the Author**

1. If the authors have adequately addressed your comments raised in a previous round of review and you feel that this manuscript is now acceptable for publication, you may indicate that here to bypass the “Comments to the Author” section, enter your conflict of interest statement in the “Confidential to Editor” section, and submit your "Accept" recommendation.

Reviewer #4: (No Response)

2. Is the manuscript technically sound, and do the data support the conclusions?

Reviewer #4: Yes

3. Has the statistical analysis been performed appropriately and rigorously? 

Reviewer #4: Yes

4. Have the authors made all data underlying the findings in their manuscript fully available?

Reviewer #4: Yes

5. Is the manuscript presented in an intelligible fashion and written in standard English?

Reviewer #4: Yes

6. Review Comments to the Author

Reviewer #4: In the previous review, I wrote, “all my comments are suggestive, not mandatory.” The authors seem to take only a few of them. Thus, there remains few I comment anymore. I’m fine; it is the editor that decides if the revision is satisfactory. That said, I dare to repeat: “I do not understand why the paper calculates ‘the [relative] homogenized cumulative incidence’ as they do. Why does it multiply the cumulative incidence by the positivity rate? And then why does it divide it by a baseline?” I wonder if a typical reader of the journal understands the term “homogenized cumulative incidence.” By the way, the manuscript changes CI_{hom, j} to r_{CI_{hom, j}}, whose subscript has the second-level subscript. It is not easy to read, and I recommend simplifying it.

Some typos?

Abstract: Relative homogenized monthly cumulative incidence ( )

line 257: rregard

7. PLOS authors have the option to publish the peer review history of their article (what does this mean?). If published, this will include your full peer review and any attached files.

Reviewer #4: No

---

## [Editor Report · Acceptance letter]

8 Feb 2022

PONE-D-21-21706R3 

Individual prevention and containment measures in schools in Catalonia, Spain, and community transmission of SARS-CoV-2 after school re-opening 

Dear Dr. Bassat:

I'm pleased to inform you that your manuscript has been deemed suitable for publication in PLOS ONE. Congratulations! Your manuscript is now with our production department. 

Kind regards, 

on behalf of

Prof. A. M. Abd El-Aty 

Academic Editor

PLOS ONE